# Identification of scavenger receptor B1 as the airway microfold cell receptor for *Mycobacterium tuberculosis*

**Haaris S Khan[1], Vidhya R Nair[1], Cody R Ruhl[1], Samuel Alvarez-Arguedas[1], Jorge L Galvan Rendiz[1], Luis H Franco[1†], Linzhang Huang[2], Philip W Shaul[2], Jiwoong Kim[3], Yang Xie[3,4,5], Ron B Mitchell[6], Michael U Shiloh[1,7]\***

[1]Department of Internal Medicine, University of Texas Southwestern Medical Center, Dallas, United States; [2]Center for Pulmonary and Vascular Biology, Department of Pediatrics, University of Texas Southwestern Medical Center, Dallas, United States; [3]Quantitative Biomedical Research Center, Department of Population and Data Sciences, University of Texas Southwestern Medical Center, Dallas, United States; [4]Harold C Simmons Cancer Center, University of Texas Southwestern Medical Center, Dallas, United States; [5]Department of Bioinformatics, University of Texas Southwestern Medical Center, Dallas, United States; [6]Department of Otolaryngology, University of Texas Southwestern Medical Center, Dallas, United States; [7]Department of Microbiology, University of Texas Southwestern Medical Center, Dallas, United States

**\*For correspondence:**
michael.shiloh@utsouthwestern.edu

**Present address:** †Federal University of Minas Gerais, Belo Horizonte, Brazil

**Competing interests:** The authors declare that no competing interests exist.

**Abstract** *Mycobacterium tuberculosis* (Mtb) can enter the body through multiple routes, including via specialized transcytotic cells called microfold cells (M cell). However, the mechanistic basis for M cell entry remains undefined. Here, we show that M cell transcytosis depends on the Mtb Type VII secretion machine and its major virulence factor EsxA. We identify scavenger receptor B1 (SR-B1) as an EsxA receptor on airway M cells. SR-B1 is required for Mtb binding to and translocation across M cells in mouse and human tissue. Together, our data demonstrate a previously undescribed role for Mtb EsxA in mucosal invasion and identify SR-B1 as the airway M cell receptor for Mtb.

## Introduction

*Mycobacterium tuberculosis* (Mtb), the causative agent of tuberculosis (TB), latently infects roughly one-third of the world's population and causes 1–2 million deaths per year. The current paradigm of acute infection is that after an actively infected person aerosolizes infectious Mtb-containing particles, a naive individual inhales the bacteria that then traverse the respiratory tree to ultimately be phagocytosed by alveolar macrophages (*Churchyard et al., 2017*; *Cohen et al., 2018*). While this model can account for pulmonary TB, it is insufficient to explain some extrapulmonary forms of TB initiated by oropharyngeal infection and lacking evidence of concurrent pulmonary disease. For example, a disease known as tuberculous cervical lymphadenopathy, or scrofula, represents 10% of all new cases of TB, and frequently manifests without lung involvement (*Fontanilla et al., 2011*). Because the oropharynx and upper airway lymphatics drain to the cervical lymph nodes, while the lower airway lymphatics drain to the mediastinal lymph nodes, infection of the cervical lymph nodes by Mtb may not involve the lower airways. Indeed, in the infamous 'Lübeck Disaster' where hundreds of infants and children were accidentally orally administered Mtb instead of the attenuated BCG

vaccine, the majority developed lymphatic and oropharyngeal TB rather than pulmonary TB (*Fox et al., 2016*), highlighting how inoculation via the oropharyngeal route can cause extrapulmonary disease.

One potential explanation for the development of lymphatic TB centers upon the mucosa-associated lymphoid tissue (MALT) (*Brandtzaeg et al., 2008*). Specialized epithelial cells known as microfold cells (M cells) overlie the MALT and are able to translocate luminal material to basolateral antigen-presenting cells located immediately beneath the M cell (*Kimura, 2018*). In this way, M cells can initiate an immune response to pathogens or material found within the lumen (*Nakamura et al., 2018*).

Since their initial discovery overlying Peyer's patches of the gastrointestinal tract, M cells have been identified at other mucosal sites. Within the respiratory tract, M cells have been found in the upper and lower airways of both mice and humans (*Fujimura, 2000*; *Kim et al., 2011*; *Kimura et al., 2019*). M cells express a number of pattern recognition receptors (PRRs) (*Mabbott et al., 2013*). The majority of these M cell receptors have been identified on gastrointestinal M cells, while receptor expression by airway microfold cells is less well understood. Some PRRs on gastrointestinal M cells function in bacterial recognition and translocation. For example, the cellular prion protein (PrP (C)), a receptor for *Brucella abortus*, is necessary for *B. abortus* translocation (*Nakato et al., 2012*). Similarly, glycoprotein 2 (GP2) expressed on the apical surface of gastrointestinal M cells recognizes FimH, a component of the type I pili found on both commensal and pathogenic bacteria (*Hase et al., 2009*). Loss of either the host receptor GP2 or the bacterial ligand FimH diminishes bacterial translocation through M cells, reducing the immune response to these antigens and bacteria within Peyer's patches (*Hase et al., 2009*).

We previously demonstrated that Mtb uses airway M cells as a portal of entry to initiate infection (*Nair et al., 2016*). We hypothesized that Mtb may produce a bacterial effector to mediate this process, and that, similar to receptors for gram-negative bacteria in the gastrointestinal tract (*Hase et al., 2009*), airway M cells may also encode an Mtb receptor. Here, we show that Mtb requires the type VII secretion system for translocation in vitro and in vivo. The type VII secretion system effector EsxA (also known as ESAT-6) is sufficient to mediate this process in vitro through binding to scavenger receptor class B type I (SR-B1). SR-B1 is enriched on mouse and human M cells both in vitro and in vivo. Loss of SR-B1 reduces EsxA and Mtb binding to M cells, and prevents Mtb translocation through M cells in vitro. Using a newly developed explanted human adenoid model, we demonstrate robust expression of SR-B1 on primary human M cells. Finally, we show that Mtb infects primary human M cells on adenoids in a type VII secretion system dependent manner. Taken together, our findings indicate that the interaction of Mtb EsxA with M cell SR-B1 allows Mtb to traverse the airway mucosa to initiate infection.

## Results

### The Mtb type VII secretion system mediates Mtb binding to and translocation through M cells in vitro

Mtb encodes several protein secretion systems important for bacterial virulence (*Feltcher et al., 2010*). One of the type VII secretion systems (T7SS) of Mtb, contained within the region of difference 1 (RD1) locus of Mtb (*Behr et al., 1999*), secretes virulence factors including EsxA and EsxB (also known as CFP-10) (*Stanley et al., 2003*; *Figure 1A*). We hypothesized that the Mtb T7SS might facilitate M cell translocation because the T7SS machine interacts directly with eukaryotic membranes (*Augenstreich et al., 2017*), EsxA can be identified on the mycobacterial cell surface (*Kinhikar et al., 2010*), and EsxA may directly bind several cell surface receptors (*Kinhikar et al., 2010*; *Sreejit et al., 2014*). To test if the Mtb T7SS was required for bacterial binding to and translocation across M cells, we used a human airway M cell transwell model that we developed previously (*Figure 1B*; *Nair et al., 2016*). Such transwells combine 16-HBE cells (*Cozens et al., 1994*) in the apical compartment of a transwell and Raji B cells in the basolateral compartment in order to mimic the organization of MALT and to enhance M cell formation (*Kernéis et al., 1997*; *Nair et al., 2016*). HBE cells cultured alone form a homogenous, polarized monolayer (hereafter called 'control') while coculture with Raji B cells induces some HBE cells to differentiate into M cells (hereafter called 'HBE/ Raji B') (*Nair et al., 2016*). We used the Mtb *eccD1::Tn5370* strain (hereafter designated Mtb

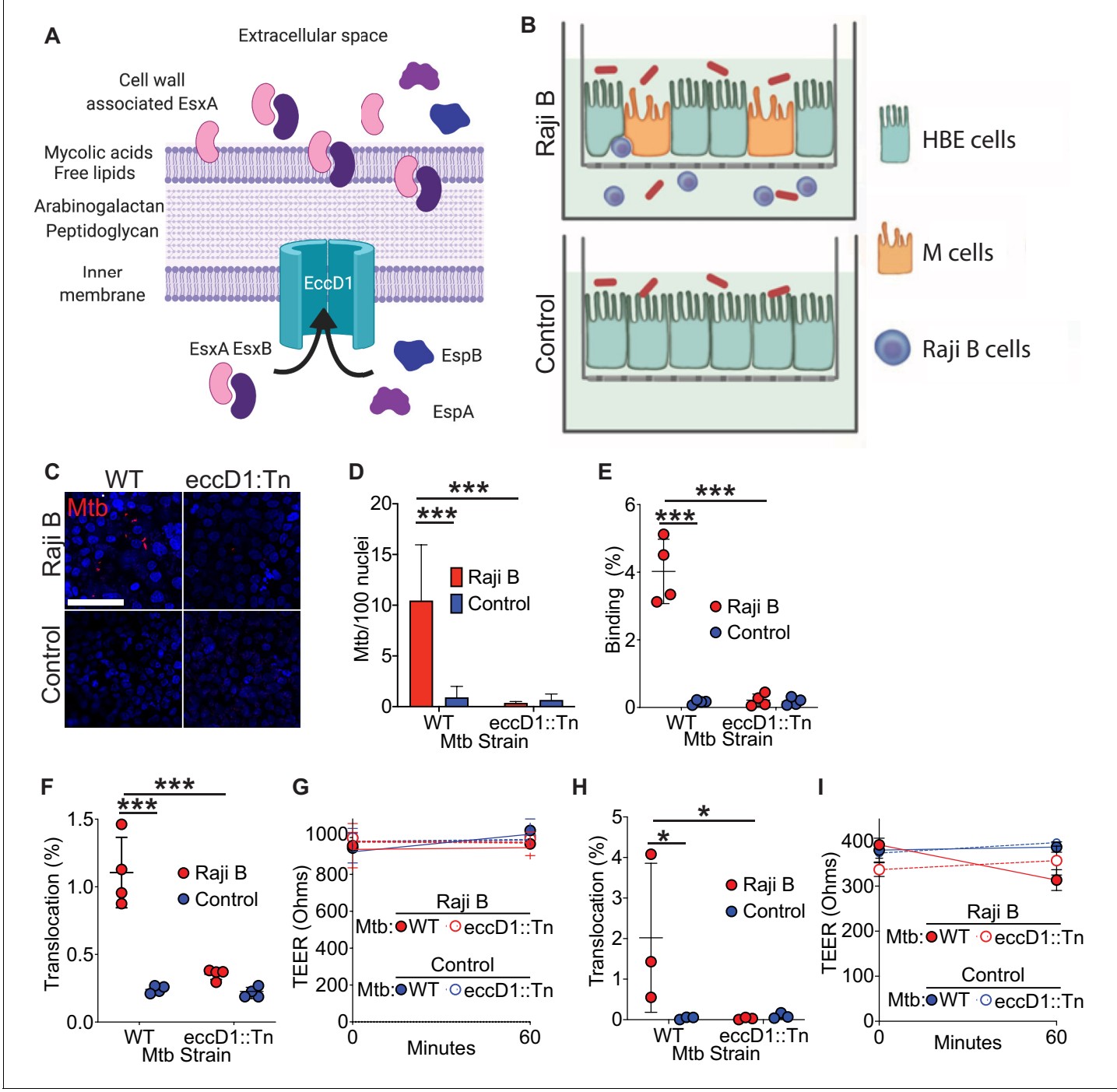

**Figure 1.** Mtb T7SS is necessary to mediate binding and translocation across M cells. (**A**) Model of Mtb T7SS. (**B**) Schematic of human airway M cell transwell model. (**C,D**) Control and HBE/RajiB transwells were incubated with Mtb strains at 4°C for 1 hr and binding was analyzed by confocal microscopy (**C**) with quantification of bacterial number (**D**). Scale bar, 20 μm. (**E**) Control and HBE/RajiB transwells were incubated with Mtb strains at 4°C for 1 hr and lysed to determine binding by quantifying bacterial CFU and comparing with the initial inoculum. (**F**) Control and HBE/RajiB transwells were incubated with Mtb strains at 37°C for 1 hr and bacterial translocation was determined by quantifying bacterial CFU from the basal compartment and comparing with the inoculum. (**G**) TEER measurements from transwells from (**F**). (**H**) Caco-2/Raji B transwells were infected as described in F. (**I**) TEER measurements from transwells from (**H**). Experiments shown are representative of at least three independent experiments. *p<0.05, ***p<0.0005 as determined by one-way ANOVA. Where not shown, comparisons were not significant.

*eccD1::Tn*) (*Cox et al., 1999*; *Stanley et al., 2003*), which lacks the inner membrane pore required for assembly of and protein secretion by the T7SS (*Figure 1A*; *Abdallah et al., 2007*). To test if the Mtb T7SS was necessary for M cell binding, we incubated transwells with either wild-type Mtb expressing mCherry (WT Mtb) or Mtb *eccD1::Tn* expressing mCherry at 4°C to prevent bacterial entry or translocation and analyzed surface binding by confocal microscopy and quantification of colony-forming units (CFU) (*Figure 1C–E*). Consistent with our prior data (*Nair et al., 2016*), significantly more WT Mtb bound HBE/Raji B transwells (containing M cells) than control transwells by both microscopy (*Figure 1C,D*) and CFU (*Figure 1E*). However, binding was greatly reduced for the Mtb *eccD1::Tn* strain (*Figure 1C–E*). To confirm that the Mtb *eccD1::Tn* strain did not harbor additional mutations other than the expected transposon insertion into *eccD1* to potentially explain this result, we performed whole genome sequencing on the Mtb Erdman WT lab strain from our lab, along with the Mtb *eccD1::Tn* strain. No significant insertions, deletions or mutations were observed in the Mtb *eccD1::Tn* strain relative to the wild-type strain, although several single-nucleotide polymorphisms were noted when we compared the strains to the reference strain (*Miyoshi-Akiyama et al., 2012*; *Supplementary file 1*).

To test if the Mtb T7SS is necessary to facilitate mycobacterial translocation across M cells, we infected the apical chamber of transwells with either WT Mtb or Mtb *eccD1::Tn* and measured translocation to the basal compartment. As we reported previously (*Nair et al., 2016*), WT Mtb translocated across HBE/Raji B transwells to a greater extent than control transwells, while the translocation of the Mtb *eccD1::Tn* strain was significantly reduced (*Figure 1F*). Importantly, the transepithelial electrical resistance (TEER), a measure of epithelial monolayer integrity (*Srinivasan et al., 2015*), was stable during the experiment (*Figure 1G*). To further verify that Mtb T7SS is required for bacterial translocation across M cells in vitro, we also utilized an established model of M cell differentiation where Caco-2 cells, a human colonic epithelial cell line, are cultured with Raji B transwells to induce M cell differentiation (*Nair et al., 2016*). Similar to HBE/Raji B transwells, we observed that Mtb translocated across Caco-2/Raji B transwells in a T7SS-dependent manner (*Figure 1H,I*). Taken together, these data show that the Mtb T7SS is necessary for both binding to and translocation across M cells in vitro.

## EsxA is sufficient to mediate binding to and translocation across M cells in vitro

Two of the most abundant T7SS secreted proteins are EsxA and EsxB (*Berthet et al., 1998*; *Brodin et al., 2005*; *Champion et al., 2014*); therefore, we hypothesized that one of these proteins might mediate Mtb binding and translocation. We expressed EsxA and EsxB as 6-His tagged constructs and purified the proteins from *E. coli* (*Figure 2—figure supplement 1*). We then conjugated recombinant EsxA, EsxB, or glycine (as a control) to fluorescent beads, added the beads to the apical chamber of transwells, and quantified bead translocation to the basal compartment by flow cytometry. EsxA-beads but not control beads translocated across HBE/Raji B transwells (*Figure 2A*) without disrupting the epithelial monolayer (*Figure 2B*). EsxA-beads, but not EsxB-beads or control beads, also translocated across Caco-2/Raji B transwells (*Figure 2C*).

To test if the ability of EsxA to mediate translocation was due to direct EsxA binding to M cells, we incubated transwells with recombinant 6xHis-tagged EsxA or EsxB and performed immunofluorescence microscopy using antibodies against 6x-His and α1,2-fucose (NKM 16-2-4; a marker for M cells [*Nair et al., 2016*; *Nochi et al., 2007*; *Figure 2D*]). While both groups of transwells had equal number of nuclei per field (*Figure 2E*), HBE/Raji B transwells had more NKM 16-2-4-positive M cells compared to control transwells (*Figure 2F*). We detected robust EsxA binding to M cells on HBE/Raji B transwells, in contrast to EsxB, which did not demonstrate significant binding (*Figure 2D,G–H*). Similar results were observed using an antibody against Sialyl Lewis[A] (SLA), a different M cell marker (*Giannasca et al., 1999*; *Nair et al., 2016*; *Figure 2—figure supplement 2*). Taken together, these data demonstrate that EsxA, but not EsxB, directly binds the M cell surface and is sufficient to mediate translocation across M cells when conjugated to inert beads.

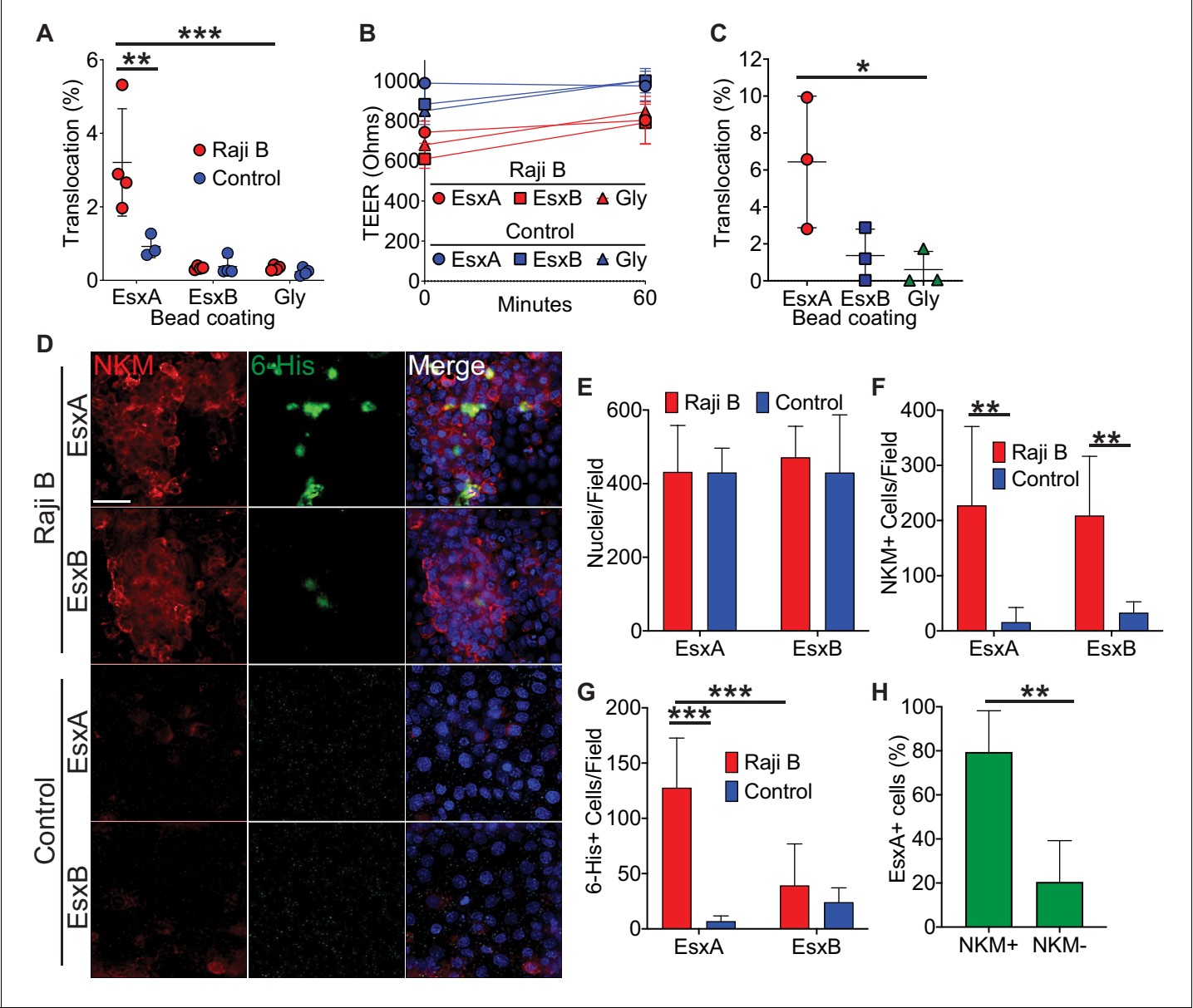

**Figure 2.** Mtb EsxA is sufficient to mediate binding and translocation across M cells. (A) Control and HBE/RajiB transwells were incubated with fluorescent beads coated with EsxA, EsxB, or glycine. Translocation was determined by comparing the number of beads in the basal compartment with the inoculum. (B) TEER measurements from transwells from (A). (C) Caco-2/Raji B transwells were treated as described in A. (D) Control and HBE/RajiB transwells were incubated with recombinant EsxA or EsxB and stained with NKM 16-2-4 (red) and an anti-6-His antibody (green). Scale bar, 30 µm. (E–G) Quantification of nuclei (E), NKM 16-2-4+ cells (F), and EsxA+ (G) cells from the transwells described in (D). (H) Quantification of NKM 16-2-4 staining on EsxA+ cells from the HBE/Raji B transwells described in (D). Experiments shown are representative of at least three independent experiments. *p<0.05, **p<0.005, ***p<0.0005 as determined by one-way ANOVA. Where not shown, comparisons were not significant.

The online version of this article includes the following figure supplement(s) for figure 2:

**Figure supplement 1.** Coommassie stain of EsxA and EsxB purification.

**Figure supplement 2.** Recombinant EsxA and SR-B1 colocalize with the M cell marker Sialyl Lewis[A] on Raji B-treated transwells.

## Scavenger receptor class B type one binds EsxA and is expressed on M cells in vitro

Because EsxA bound directly to the surface of M cells, we hypothesized that EsxA may engage a cell surface receptor. To affinity purify cell surface binding proteins, we performed a modified co-immunoprecipitation experiment using either EsxA or transferrin crosslinked to the TriCEPS reagent, a

molecule that allows for the covalent cross-linking of a ligand and its receptor (*Tremblay and Hill, 2017*). We used Caco-2 cells for this experiment as they have been used extensively as a model for M cells in vitro (*Tyrer et al., 2006*) and because Caco-2/Raji B transwells behaved similarly to HBE/Raji B transwells in Mtb and EsxA translocation (*Figure 1E,G* and *Figure 2A,C*). Using this approach, we identified the interaction between transferrin (TRFE) and the transferrin receptor (TFR1) (*Figure 3A*, blue peptides; *Supplementary file 2*), proving the validity of this system. When cells were treated with EsxA, peptides for two proteins, apolipoprotein E (ApoE) and scavenger receptor class B type I (SR-B1) were enriched (*Figure 3A*, red peptides; *Supplementary file 2*). Because ApoE is a soluble protein (*Huang and Mahley, 2014*) while SR-B1 is a known cell surface molecule, we focused on SR-B1. For further verification, we performed a co-immunoprecipation/biotin transfer experiment without the TriCEPS reagent. After incubation with biotinylated EsxA, completion of the biotin transfer assay, and subsequent immunoprecipitation with streptavidin-coated beads, western blotting with an anti-SR-B1 antibody detected an approximately 82 kD band consistent with the known observed molecular weight of glycosylated SR-B1 (*Acton et al., 1996*; *Figure 3B*).

We next determined if SR-B1 expression is specific for M cells or ubiquitously expressed by epithelial cells. We quantified colocalization of SR-B1 and NKM 16-2-4 by immunofluorescence microscopy in control and HBE/Raji B transwells (*Figure 3C–F*). While there was no difference in the number of nuclei per field on the transwells (*Figure 3C*), SR-B1 expression was higher on HBE/Raji B transwells (*Figure 3C,E*) and the majority of the SR-B1 positive cells were NKM[+] M cells (*Figure 3C, F*). Taken together, we identify SR-B1 as a candidate EsxA receptor expressed on M cells in vitro.

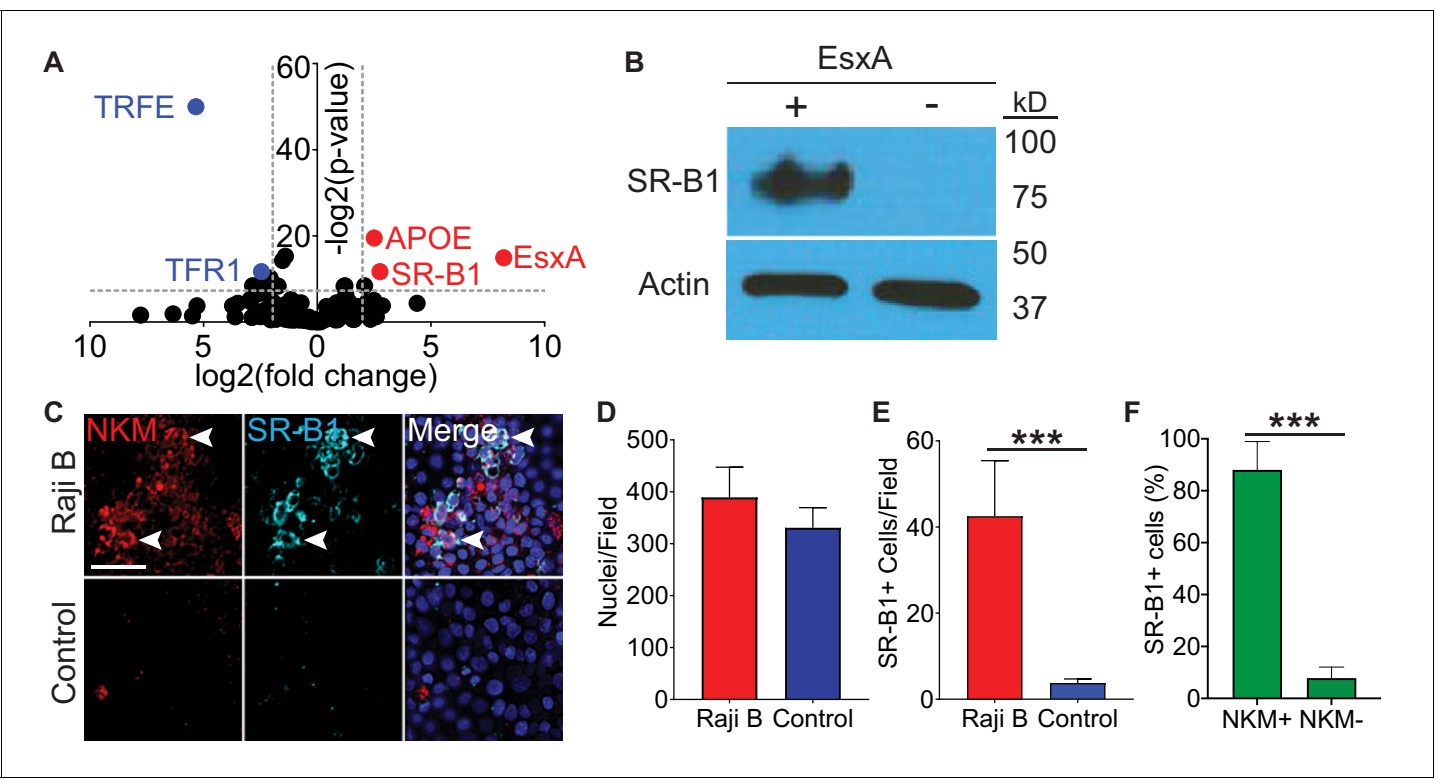

**Figure 3.** SR-B1 is the M cell EsxA receptor. (A) Volcano plot displaying peptides enriched when Caco-2 cells were treated either with transferrin (blue dots on left) or EsxA (red dots on right). Results are of a single experiment with three biologic replicates per condition. (B) Western blot using an anti-SR-B1 antibody (top) or an anti-actin antibody (bottom) of proteins enriched after HBE cells were incubated with biotinylated EsxA or control. (C) Control and HBE/RajiB transwells were stained with NKM 16-2-4 and an anti-SR-B1 antibody and analyzed by confocal microscopy. Arrows denote examples of double positive cells. Scale bar, 40 μm. (D,E) Quantification of the number of nuclei (D) or SR-B1[+] cells (E) from the transwells described in (C). (F) Quantification of NKM 16-2-4 staining on SR-B1[+] cells from the HBE/Raji B transwells described in (C). Experiments shown are representative of at least three independent experiments. ***p<0.0005 as determined by one-way ANOVA. Where not shown, comparisons were not significant.

## Genetic disruption of SR-B1 limits EsxA binding to M cells

We next investigated whether SR-B1 is required for EsxA binding to M cells by exposing cells to recombinant EsxA in the presence or absence of SR-B1. We first transduced HBE cells with non-targeting (NT) or *SR-B1* shRNA and observed a robust knock-down of SR-B1 in HBE cells transduced with the *SR-B1* shRNA as compared to the NT shRNA (*Figure 4A*). HBE/Raji B transwells constructed from these cells had a similar number of M cells comparing NT and *SR-B1* shRNA transwells (*Figure 4B,C*) and *SR-B1* shRNA transwells showed a reduction in SR-B1 expression by immunofluorescence microscopy (*Figure 4B,D*). When we incubated these transwells with recombinant EsxA, absence of SR-B1 reduced the number of EsxA positive cells on HBE/Raji B *SR-B1* shRNA transwells (*Figure 4B,E*). Additionally in HBE/RajiB NT shRNA transwells, the majority of EsxA-positive cells were SR-B1 positive (*Figure 4B,F*), suggesting that EsxA preferentially bound SR-B1 expressing M cells. Taken together, we identify SR-B1 as necessary for EsxA binding to M cells.

## Genetic disruption of SR-B1 reduces both Mtb binding to and translocation across M cells

To determine the role of SR-B1 in Mtb binding to M cells, we incubated HBE/Raji B transwells expressing NT or *SR-B1* shRNAs with mCherry Mtb at 4°C for 1 hr and analyzed binding by confocal microscopy and CFU (*Figure 4G–I*). Loss of SR-B1 greatly reduced the number of bacteria bound to the HBE/Raji B transwells as determined by confocal microscopy (*Figure 4G,H*) and by quantification of CFU (*Figure 4I*). To determine the role of SR-B1 in Mtb translocation by M cells, we infected HBE/Raji B transwells expressing NT or *SR-B1* shRNAs with Mtb in the apical compartment and measured translocation to the basal compartment. As expected from the reduced bacterial binding (*Figure 4G–I*), loss of SR-B1 also greatly reduced Mtb translocation in the HBE/Raji B transwells at 37°C (*Figure 4J*) with no impact on the TEER (*Figure 4K*). The reduced ability of Mtb to translocate across HBE/Raji B transwells expressing *SR-B1* shRNA was not due to any intrinsic defect in translocation caused by SR-B1 deficiency as another airway pathogen, *Pseudomonas aeruginosa*, was able to translocate equally across NT and *SR-B1* shRNA HBE/Raji B transwells (*Figure 4L*). Of note, *P. aeruginosa* does not encode a T7SS or EsxA homologue, suggesting that its translocation across M cells depends on unique bacterial and host factors. We thus conclude that SR-B1 is essential for the binding and translocation of Mtb via M cells in a process requiring the effector EsxA.

## The Mtb type VII secretion system is necessary for Mtb translocation in mice

M cells are found in the upper and lower airways of mice and humans (*Fujimura, 2000*; *Mutoh et al., 2016*; *Nair et al., 2016*; *Teitelbaum et al., 1999*). We therefore determined if SR-B1 was expressed preferentially by primary M cells as compared to other epithelial cells using immunofluorescence microscopy. Mouse nasal-associated lymphoid tissue (NALT), a region enriched for M cells (*Mutoh et al., 2016*; *Nair et al., 2016*; *Park et al., 2003*), demonstrated robust SR-B1 staining on the surface of NKM 16-2-4 positive cells (*Figure 5A*). Importantly, we did not observe SR-B1$^+$/NKM 16-2-4$^-$ cells, demonstrating that SR-B1 is specific for M cells in the NALT epithelia in vivo.

We and others previously demonstrated that NALT and airway M cells are a portal of entry for Mtb in mice (*Nair et al., 2016*; *Teitelbaum et al., 1999*). To determine if the Mtb T7SS is necessary for bacterial translocation in vivo, we performed NALT infections (*Nair et al., 2016*) with Mtb *eccD1::Tn* or Mtb*ΔesxA*. As described above for Mtb *eccD1::Tn*, to control for possible mutations outside the known *esxA* deletion, we performed whole genome sequencing on the Mtb*ΔesxA* strain and determined that its sequence is essentially identical to the parental Mtb WT strain (*Supplementary file 1*). Of note, in both Mtb mutant strains the T7SS machine fails to assemble (*Abdallah et al., 2007*) thereby preventing T7SS-dependent virulence factor secretion. We infected mice intranasally with WT Mtb, Mtb *eccD1::Tn* (*Figure 5B*), or Mtb*ΔesxA* (*Figure 5C*) and enumerated CFU from draining cervical lymph nodes 7 days post-infection (*Nair et al., 2016*). Both the Mtb *eccD1::Tn* and Mtb*ΔesxA* strains had 1.0–1.5 log fewer bacteria compared to WT Mtb in the cervical lymph nodes. This degree of attenuation was not observed when we infected mice intranasally with a Cor-deficient strain of Mtb (Mtb*cor::Tn7*) that is also attenuated in vivo (*Zacharia et al., 2013*; *Figure 5—figure supplement 1*). Taken together, these data demonstrate that the lower CFU

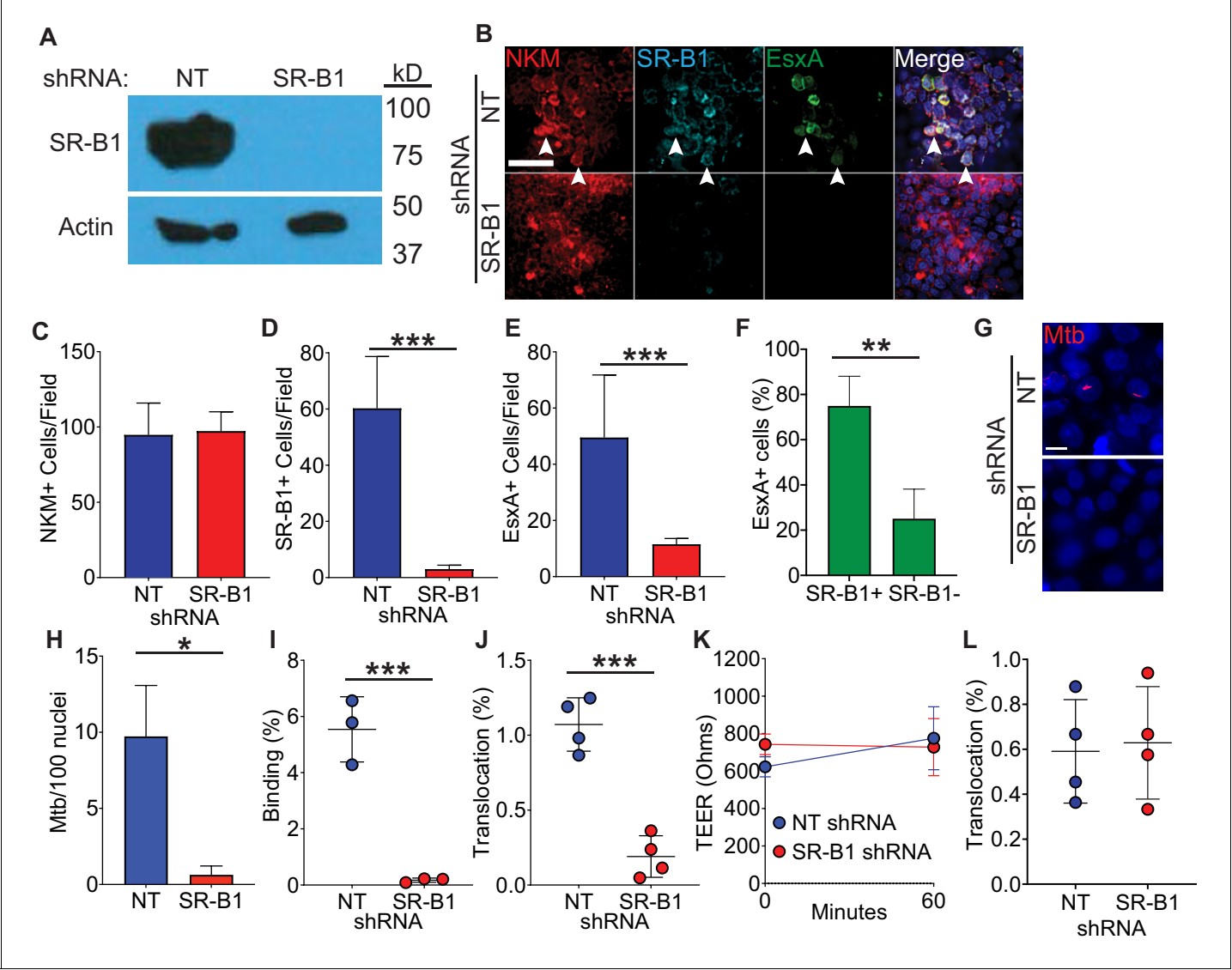

**Figure 4.** Loss of SR-B1 reduces EsxA binding and Mtb translocation through M cells. (A) Western blot of SR-B1 (top) or beta-actin (bottom) of shRNA expressing HBE cells. (B) HBE/RajiB transwells with shRNA expressing HBE cells were incubated with biotinylated EsxA and stained with NKM 16-2-4 (red), anti-SR-B1 (cyan), and Alexa Fluor 488 conjugated streptavidin (green). Arrows denote examples of triple positive cells. Scale bar, 40 µm. (C–E) Quantification of the number of NKM[+] (C), SR-B1[+] cells (D) and EsxA[+] cells (E) on transwells described from (B). (F) Quantification of SR-B1 staining on EsxA[+] cells from HBE NT shRNA/Raji B transwells described in (B). (G,H) HBE/RajiB transwells with shRNA expressing HBE cells were incubated with mCherry Mtb and Mtb binding was analyzed by confocal microscopy (G) with quantification of bacterial number (H). Scale bar, 10 µm. (I) HBE/RajiB transwells with shRNA expressing HBE cells were incubated with Mtb strains at 4°C and lysed to determine binding by quantifying bacterial CFU and comparing with the initial inoculum. (J) HBE/RajiB transwells with shRNA expressing HBE cells were incubated with Mtb strains at 37°C and bacterial translocation was determined by quantifying bacterial CFU from the basal compartment and comparing with the inoculum. (K) TEER of the transwells from (J). (L) HBE/RajiB transwells with shRNA expressing HBE cells were incubated with *Pseudomonas aeruginosa* at 37°C and bacterial translocation was determined by quantifying bacterial CFU from the basal compartment and comparing with the inoculum. Experiments shown are representative of at least three independent experiments. *p<0.05, **p<0.005, ***p<0.0005 as determined by one-way ANOVA. Where not shown, comparisons were not significant.

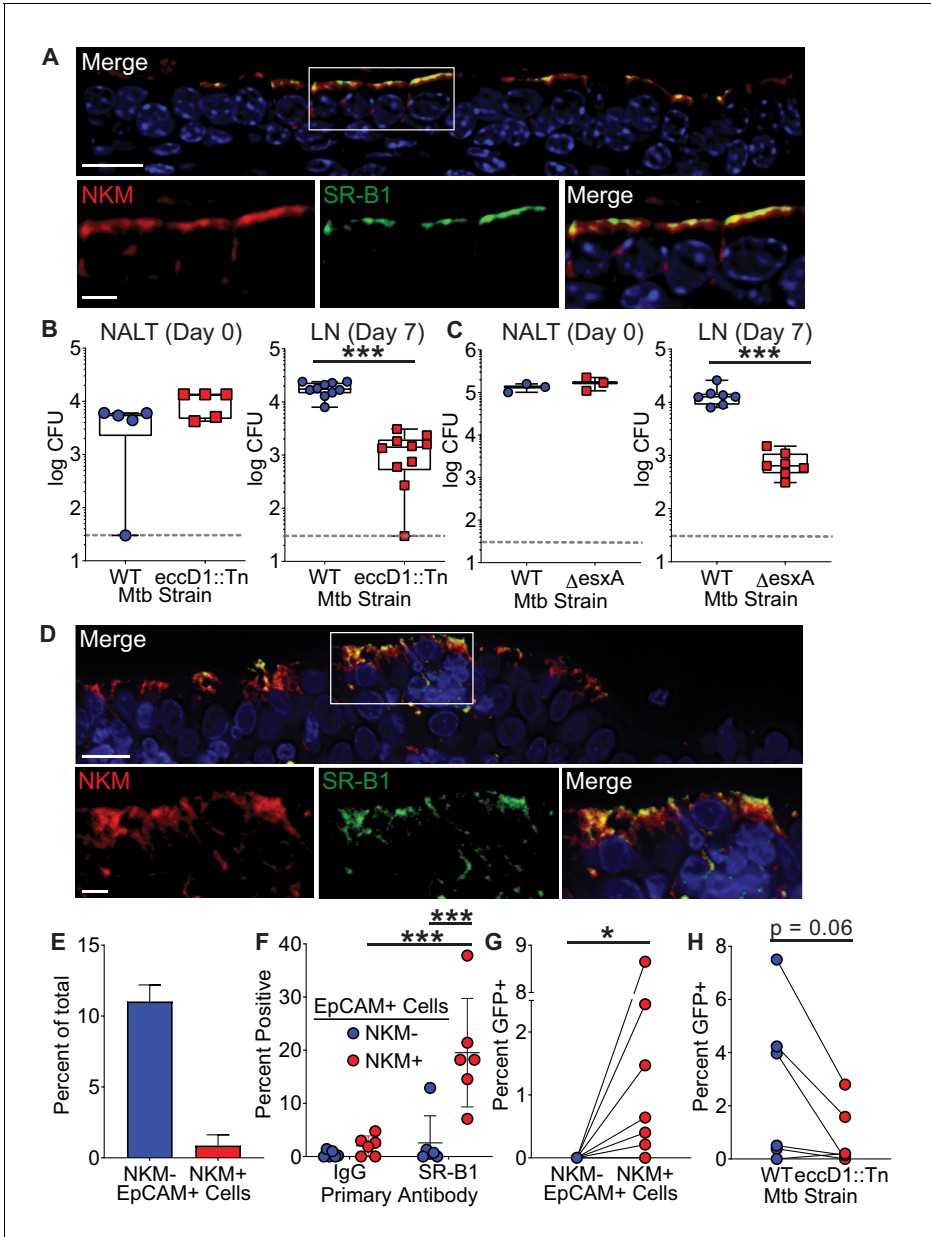

**Figure 5.** The Mtb type VII secretion system is necessary for Mtb entry in mice and humans. (**A**) Mouse NALT sections were stained with NKM 16-2-4 and anti-SR-B1 antibodies and analyzed by confocal microscopy. Scale bar, top, 15 μm, bottom, 5 μm. (**B,C**) Mice were intranasally infected with either WT Mtb, Mtb *eccD1*::Tn (**B**), or Mtb*ΔesxA* (**C**). CFU was determined in the NALT on day 0 (left) or in the cervical lymph nodes on day 7 (right). Symbols represent CFU from individual animals (n = 8–10 per strain). ***p<0.0005 compared to WT by Mann-Whitney U test. (**D**) Human adenoid sections were stained with NKM 16-2-4 and anti-SR-B1 antibodies and analyzed by confocal microscopy. Scale bar, top, 15 μm, bottom, 5 μm. (**E**) Human adenoids were disaggregated, stained with NKM 16-2-4 and anti-EpCAM antibodies, and analyzed by flow cytometry. (**F**) Human adenoids were treated as in (**E**), stained with anti-SR-B1 or control IgG antibodies and analyzed by flow cytometry. Symbols represent adenoids from individual donor (**F–H**). **p<0.005, Wilcoxon matched pairs signed rank test. (**G**) Human adenoids were infected with GFP+ Mtb, disaggregated, immunostained and analyzed by flow cytometry to determine the proportion of GFP+ Mtb containing NKM+/EpCAM+ and NKM−/EpCAM+ cells. *p<0.05, Wilcoxon matched pairs signed rank test. (**H**) Human adenoids were infected with GFP+ Mtb or GFP+ Mtb *eccD1*::Tn. The percentage of GFP+ Mtb containing NKM+/EpCAM+ double positive cells was determined by flow cytometry. The Wilcoxon matched pairs signed rank test was used for comparison.

The online version of this article includes the following figure supplement(s) for figure 5:

**Figure supplement 1.** Mtbcor::Tn7 does not display a translocation defect following a mouse intranasal infection Mice were intranasally infected with either WT Mtb or Mtbcor::Tn7.

**Figure supplement 2.** Adenoid gating strategy to determine SR-B1 positive cells.

**Figure supplement 3.** Adenoid gating strategy to determine GFP+ Mtb containing cells.

recovered from cervical lymph nodes of mice infected with T7SS mutant Mtb strains may be due to reduced translocation across M cells.

## Human adenoid M cells express SR-B1

Because TB is a human disease, we tested if primary human M cells can serve as a portal of entry for Mtb. The human adenoid is a MALT structure that contains M cells interspersed among the overlying epithelial cells (*Fujimura, 2000*). We first demonstrated that human adenoids contain SR-B1$^+$/NKM$^+$ M cells by immunofluorescence microscopy (*Figure 5D*), similar to our observations from mouse NALT (*Figure 5A*). We next confirmed the presence of M cells in human adenoids by flow cytometry (*Figure 5—figure supplement 2* for gating strategy) and observed that approximately 10% of adenoid cells were EpCAM$^+$/NKM$^-$ epithelial cells, while about 1% of the cells were EpCAM$^+$/NKM$^+$ double positive M cells (*Figure 5E*). When we analyzed SR-B1 expression using flow cytometry, we observed that approximately 20% of primary human M cells (marked as EpCAM$^+$/NKM$^+$ cells) were SR-B1 positive as compared to less than 2% of the other epithelial cells (EpCAM$^+$/NKM$^-$) (*Figure 5F*), verifying our observation that SR-B1 is expressed predominately by M cells in vivo.

## Human adenoid M cells are a portal of entry for Mtb

To determine if primary human M cells can be a route of entry for Mtb, we infected human adenoids with GFP$^+$ Mtb and quantified the number of GFP$^+$ Mtb EpCAM$^+$/NKM$^+$ M cells versus GFP$^+$ Mtb EpCAM$^+$/NKM$^-$ epithelial cells by flow cytometry (*Figure 5G*, *Figure 5—figure supplement 3* for gating strategy). The number of GFP$^+$ Mtb containing EpCAM$^+$/NKM$^+$ M cells ranged from 0.1–8% while we were unable to identify any GFP$^+$ Mtb containing EpCAM$^+$/NKM$^-$ epithelial cells (*Figure 5G*), suggesting that M cells were a preferred route of entry for Mtb in adenoids. Finally, we determined the role of the Mtb T7SS in Mtb entry into adenoid M cells. We infected adenoids with GFP$^+$ WT Mtb or GFP$^+$ Mtb *eccD1::Tn* and observed more GFP$^+$/EpCAM$^+$/NKM$^+$ cells after infection with WT Mtb as compared to Mtb *eccD1::Tn* (*Figure 5H*). Taken together, we conclude that Mtb can enter via mouse NALT and human adenoid M cells in a T7SS dependent manner.

## Discussion

In this work, we used in vitro and in vivo M cell models to demonstrate a mucosal interaction between Mtb EsxA and the cell surface protein SR-B1. EsxA, a protein secreted through the T7SS, bound M cells in vitro, was sufficient to mediate M cell translocation by inert beads and was necessary for Mtb translocation in vitro. Furthermore, the T7SS was necessary for Mtb translocation in a mouse mucosal infection model. Primary human airway M cells internalized Mtb in a T7SS-dependent manner, indicating that this process is relevant for human disease. Finally, SR-B1 was enriched on M cells and served as a receptor for Mtb EsxA to mediate Mtb translocation. Together, our data demonstrate a previously undescribed role for Mtb EsxA in mucosal invasion and identify SR-B1 as the airway M cell receptor for Mtb.

EsxA has previously been implicated as a secreted pore-forming molecule (*Smith et al., 2008*), although this activity has recently been questioned (*Conrad et al., 2017*). In our experiments utilizing recombinant EsxA, we also did not observe pore formation or epithelial damage. This could be due to the relatively short amount of time we incubated EsxA with our transwells for binding or translocation experiments. Alternatively, the pore forming properties of EsxA may only occur when the protein is in low pH conditions, such as in the lysosome. Thus, EsxA may directly interact with M cell SR-B1 in a cell contact-dependent manner (*Conrad et al., 2017*), leading to SR-B1 receptor-mediated internalization similar to its function in both hepatitis C virus and *Plasmodium vivax* uptake (*Heo et al., 2006*; *Manzoni et al., 2017*). Although SR-B1 has not been previously identified as an EsxA receptor, prior studies have found other host proteins that interact with EsxA, including laminin (*Kinhikar et al., 2010*), β2 microglobulin (*Sreejit et al., 2014*), and TLR-2 (*Pathak et al., 2007*). We did not identify these proteins in our affinity purification assay, a discrepancy possibly related to the cell types used for binding experiments. However, it is possible that these or other receptors could partly compensate for loss of SR-B1 on M cells, explaining the low level of binding and translocation observed in the absence of SR-B1. Alternatively, Mtb may encode for factors other than EsxA that also mediate M cell binding and translocation, explaining the low level of binding and translocation observed in the absence of EsxA.

In mice, we observed that Mtb lacking the T7SS had a greatly reduced ability to disseminate from mouse NALT to the cervical lymph nodes, potentially due to a reduced ability to translocate across M cells. A possible alternate interpretation for this result centers on the observation that T7SS deficient strains of Mtb are attenuated in vivo and in macrophages (*Stanley et al., 2003*). Thus, the reduced CFU recovered from draining lymph nodes could represent a macrophage survival defect for the T7SS deficient strains. However, when we used a different attenuated Mtb strain for NALT infection, we observed normal dissemination to the draining lymph nodes. We therefore propose that the reduced CFU recovered from cervical lymph nodes of mice infected with T7SS-deficient Mtb is not simply due to an attenuation defect within macrophages. Consistent with this interpretation, the markedly reduced translocation of T7SS-deficient Mtb across M cells in vitro and into explanted human adenoids ex vivo, in the absence of an innate immune response and over a very short time course, indicates that the T7SS is required for translocation across M cells.

SR-B1 has been well characterized as a high-density lipoprotein receptor involved in cholesterol uptake (*Acton et al., 1996*). It has also been shown that SR-B1 binds several bacterial molecules, including lipopolysacharide and lipoteicheic acid produced by gram-negative and gram-positive bacteria, respectively (*Bocharov et al., 2004*). Although direct interaction of EsxA and SR-B1 has not previously been shown, SR-B1 has been reported as a receptor for mycobacteria (*Philips et al., 2005*; *Schäfer et al., 2009*), primarily in macrophages (reviewed in *Stamm et al., 2015*). However, when SR-B1$^{-/-}$ mice were infected with Mtb via the aerosol route, there was no difference in bacterial replication, granuloma size, cytokine secretion, or survival within the first 4 months post-infection compared to wild-type mice (*Schäfer et al., 2009*). Based on our current data and previous results showing improved mouse survival during aerosol Mtb infection when M cells are reduced (*Nair et al., 2016*), we predict that loss of M cell SR-B1 should reduce bacterial dissemination from the airway and enhance mouse survival. SR-B1$^{-/-}$ mice experience defective intrauterine and post-natal development and as a result are not born at normal Mendellian ratios (*Santander et al., 2013*). In addition, they manifest increased serum HDL, cardiovascular defects and altered adrenal hormones (*Trigatti et al., 1999*), making them incompatible with such a study. Likewise, mice expressing an M-cell specific Cre have not been reported, preventing analysis of SR-B1 function exclusively in M cells.

Adenoid M cells may serve as a portal of entry for Mtb, with significant implications for Mtb pathogenesis in humans. Because respiratory MALT is more abundant in children than adults (*Tschernig and Pabst, 2000*) and M cells are a key component of MALT (*Corr et al., 2008*), we propose that the increased incidence of extrapulmonary TB in children (*Yang et al., 2004*) is due to M cell mediated translocation. Interestingly, there was significant variation in M cell entry in human adenoids, which could relate to polymorphisms in SR-B1 or differences in SR-B1 expression by M cells.

Many bacterial and viral pathogens use the airway as a portal of entry, such as *P. aeruginosa* (*Bucior et al., 2012*), *Bacillus anthracis* (*Russell et al., 2008*), *Streptococcus pneumonia* (*Wilkosz et al., 2012*), *Streptococcus pyogenes* (*Barnett et al., 2015*), severe acute respiratory syndrome coronavirus (*Liu et al., 2016*) and varicella zoster virus (*Messaoudi et al., 2009*). Similarly, a variety of pathogens that invade via the gastrointestinal tract such as *B. abortus* via PrP(C) (*Nakato et al., 2012*), *S. typhimurium* via GP2 (*Hase et al., 2009*), and murine norovirus (*Gonzalez-Hernandez et al., 2014*) possibly via CD300lf (*Haga et al., 2016*; *Orchard et al., 2016*) use unique receptors for M cell mediated translocation. Thus, we speculate that a broad array of airway pathogens exploit distinct M cell receptors to penetrate the airway mucosa and disseminate.

In conclusion, we demonstrate that M cells are a portal of entry for Mtb in vitro, in mouse NALT, and in human adenoids. Utilizing mouse models and in vitro models, we identify EsxA and SR-B1 as a molecular synapse required for Mtb translocation across M cells in vitro and in vivo in both mice and humans. A greater understanding of the role of airway M cells in the context of infection by Mtb or other respiratory pathogens will yield insight into novel pathways with potential for new vaccine candidates or therapeutics.

## Materials and methods

### Bacterial strains and media

*M. tuberculosis* Erdman, *M. tuberculosis* Erdman *eccD1::Tn5370* (*Cox et al., 1999*; *Stanley et al., 2003*), *M. tuberculosis* Erdman *ΔesxA* (*Stanley et al., 2003*), *M. tuberculosis* Erdman *cor:Tn7* (*Zacharia et al., 2013*) were grown in Middlebrook 7H9 medium or on Middlebrook 7H11 plates supplemented with 10% oleic acid-albumin-dextrose-catalase. Tween 80 (Fisher T164-500) was added to liquid medium to a final concentration of 0.05%. Strains *M. tuberculosis* Erdman, *M. tuberculosis* Erdman *eccD1::Tn5370* and *M. tuberculosis* Erdman *ΔesxA* underwent whole genome sequencing to determine the presence of unknown genetic polymorphisms.

### Whole genome sequencing

*M. tuberculosis* Erdman, *M. tuberculosis* Erdman *eccD1::Tn5370*, *M. tuberculosis* Erdman *ΔesxA* were grown to late-log phase and genomic DNA isolated by the cetyltrimethylammonium bromide (CTAB)-lysozyme method (*Larsen et al., 2007*). Genomic DNA was then enzymatically fragmented and sequenced using Illumina NextSeq 550 sequencing (Microbial Genome Sequencing Center, https://migscenter.com). To reconstruct the bacterial genomes and identify genetic modifications, the bioinformatics analysis workflow was based on Genome Analysis Toolkit (GATK, v3.8–0; RRID: SCR_001876) (*DePristo et al., 2011*; *McKenna et al., 2010*) best practices. Quality control and adapter trimming were performed using Trim Galore (v0.6.4; RRID:SCR_011847) (https://github.com/FelixKrueger/TrimGalore). Burrows-Wheeler Aligner (BWA, v0.7.17; RRID:SCR_010910) (*Li and Durbin, 2009*) was employed to map the reads to the genome of the publicly available Mtb Erdman (ATCC35801) strain (NCBI assembly: ASM35020v1). Picard (v2.12.0; RRID:SCR_006525) (https://broadinstitute.github.io/picard) was used to remove PCR indices. Variant calling and genotyping were performed using GATK HaplotypeCaller (RRID:SCR_001876) and the variant calls were filtered by applying the following criteria: DP (Approximate read depth)<10, GQ (Genotype Quality)<20. The variants were annotated using a custom Perl script (https://github.com/jiwoongbio/Annomen). Insertions and deletions were identified using coverage depths and split reads from SAMtools (v0.1.19; RRID:SCR_002105) (*Li et al., 2009*). SPAdes (v3.13.0; RRID:SCR_000131) (*Bankevich et al., 2012*) was used to de novo assembly and MUMmer 4 (RRID:SCR_001200) (*Kurtz et al., 2004*) was used to compare the genome assemblies. The genomes are available at NCBI Sequence Read Archive Accession #PRJNA605439.

### Cell culture

The human colorectal adenocarcinoma cell line Caco-2 (HTB-37; RRID:CVCL_0025) and human Burkitt lymphoma cell line Raji B (CCL-86; RRID:CVCL_0511) were obtained from ATCC (Manassas, VA). 16HBE14o- cells (RRID:CVCL_0112) (*Forbes et al., 2003*) were provided by Dieter Gruenert (University of California, San Francisco). Caco-2 or HBE cells were grown in DMEM (Gibco 11965092) supplemented with 20% fetal bovine serum (Gibco 26140079), 50 units/mL penicillin (Gibco 15140122), 50 µg/mL streptomycin (Gibco 15140122), 2 mM L-glutamine (Gibco 25030081), 1% sodium pyruvate (Gibco 11360070), 1% non-essential amino acids (Gibco 11140050), and 1 mM HEPES (Hyclone SH30237.01). Raji B cells were grown in DMEM supplemented with 20% FBS and 2 mM L-glutamine. In order to generate stable knock-down lines of SR-B1, HBE cells were transduced with lentivirus containing the appropriate shRNA cloned into pLKO.1 (Addgene 10878) as described previously (*Huang et al., 2019*). Transduced cells were selected with puromycin (Sigma-Aldrich P8833-10MG) and surviving cells were maintained in puromycin for three additional passages. All cell lines were routinely tested for mycoplasma contamination. HBE, Caco-2 and Raji B cell lines were verified using STR profiling analysis (ATCC).

### Tissue bilayer model

$3 \times 10^5$ Caco-2 or HBE cells in 1 mL of media were plated in the upper chamber of a 3 µm transwell insert (Corning 3462). For Raji B treated transwells, $5 \times 10^5$ Raji B cells in 2 mL of media were added to the basal compartment, thereby inducing some of the overlying epithelial cells to differentiate into M cells. For control transwells, 2 mL of media alone were added to the basal chamber, leading to little to no M cell differentiation. 1 mL of media in the upper chamber and 1 mL of media in the

bottom chamber were aspirated daily and replaced with 1 mL of fresh media. The transwells were maintained at 37℃ for 2 weeks or until the transepithelial electrical resistance was greater than 350 Ω. 72 hr prior to infection, transwells were cultured in media lacking antibiotics. Transwell media was changed approximately 2 hr prior to infection.

### In vitro Mtb infection

Liquid cultures of Mtb were grown until mid-log phase, washed three times with PBS, and centrifuged and sonicated to remove clumps. Bacteria were then resuspended in DMEM + 20% fetal bovine serum. For translocation assays, bacterial inoculum was added to the upper chamber of the transwell at a MOI of 5:1 and media from the basal compartment was sampled after 60 min. The samples were then plated on 7H11 agar plates and maintained in a 37℃ incubator for 3 weeks to allow for colony formation.

### Protein expression and purification

gBlocks (IDT) encoding Mtb EsxA or EsxB were first cloned into the pENTR entry vector (Thermo K240020) then subcloned into the pDest17 destination vector (Thermo 11803012; Thermo 11791020) using Gateway cloning (Invitrogen) per the manufacturer's protocol. The resulting vectors were cloned into the BL21 strain of *E. coli* (NEB C2527I) for protein expression. 1 L of bacterial culture was grown to an OD600 of 0.6, induced with 1 mM IPTG (Promega V3955) at 37℃ for 3 hr, and centrifuged at 3500 rpm for 15 min at 4℃ to yield a bacterial pellet. The bacterial pellet was then resuspended in 15 mL of resuspension buffer (50 mM sodium phosphate, 500 mM NaCl, pH 7.4) with one tablet of EDTA-free protease inhibitor (Roche 11836170001). Bacteria were lysed by sonication and centrifuged at 11,200 rpm for 15 min at 4℃. The resulting pellet was resuspended in 20 mL of 8 M urea in resuspension buffer and incubated for 2 hr at room temperature with gentle agitation. The protein slurry was again centrifuged at 11200 rpm for 15 min at 4℃ and the resulting supernatant was incubated with cobalt TALON affinity resin (Clontech 635503) for 2 hr at room temperature. Resin was washed with 8 M urea in resuspension buffer and EsxA or EsxB was eluted with 150 mM imidazole and 8 M urea in resuspension buffer. The eluate was dialyzed overnight using a Slide-a-Lyzer dialysis cassette (Thermo 66203) against 10 mM ammonium bicarbonate. The dialyzed sample was again incubated with cobalt TALON affinity resin for 2 hr at room temperature. Resin was subsequently washed with 10 mM Tris-HCl pH 8.0, 0.5% ASB-14 (Sigma A1346-1G) in 10 mM Tris-HCl pH 8.0, and 10 mM Tris-HCl pH 8.0. EsxA or EsxB was eluted with 150 mM imidazole in PBS, dialyzed overnight against PBS, and stored at 4℃. Fractions were analyzed by SDS-PAGE followed by Coommassie staining with Brilliant Blue R-250 (Fisher BP101-25).

### Tissue bilayer immunofluorescence microscopy

In order to image binding of Mtb to transwells, mCherry Mtb was grown until mid-log phase, washed, and centrifuged and sonicated to remove clumps. The bacterial inoculum was added to the upper chamber of the transwell at a MOI of 5:1 for 2 hr at 4℃ with gentle agitation every 15 min. Transwells were gently washed and fixed with 4% paraformaldehyde in PBS at 4℃ for 1 hr. Transwell inserts were stained with DAPI (Thermo D1306), excised using a blade, mounted on microscope slides using Prolong Gold antifade reagent (Invitrogen P36390) and imaged using an AxioImager MN microscope (Zeiss). In order to image binding of EsxA to transwells, EsxA was expressed and purified as described above. EsxA was then biotinylated by the Sulfo-SBED reagent (Thermo 33033) per manufacturer's instructions and excess reagent was removed using PD-10 desalting columns (GE Healthcare 17-0851-01). Transwells were then incubated with 1.5 μM EsxA in HBSS for 2 hr at 4℃ with gentle agitation, washed, and exposed to UV light for 30 min at room temperature to allow for cross-linking. Transwells were then fixed with 4% paraformaldehyde in PBS for 15 min at room temperature, blocked with 10% donkey serum (Sigma D9663-10ML) in PBS for three hours at room temperature, and incubated with a 1:100 dilution of rabbit anti-SR-B1 antibody (Abcam 52629; RRID: AB_882458) in 2% donkey serum in PBS overnight at 4℃. The following day, transwells were washed and incubated with a 1:100 dilution of PE-conjugated rat NKM 16-2-4 (Miltenyi 130-102-150; RRID: AB_2660295), a 1:500 dilution of an AlexaFluor 647 conjugated donkey-anti-rabbit secondary antibody (Thermo A-31573; RRID:AB_2536183), and a 1:500 dilution of AlexaFluor 488 conjugated streptavidin (Jackson 016-540-084; RRID:AB_2337249) for 1 hr at room temperature. Transwells

were then washed, stained with DAPI, excised with a blade, mounted, and imaged as described above. Five fields of view were imaged per independent experiment.

## Microsphere conjugation and translocation

1 µm microspheres were conjugated to protein as per instructions (Polylink 24350–1). Briefly, 12.5 mg of microspheres were centrifuged and washed twice in coupling buffer. Microspheres were then incubated with an EDAC/coupling buffer solution to activate the microspheres. 200 µg of protein is added to the beads, thereby allowing for covalent binding of the protein to the microspheres. Microspheres are then washed twice with PBS and stored at 4°C. In order to test the ability of these beads to translocate in the tissue bilayer assay, beads were diluted to a MOI of 5:1 in DMEM + 20% fetal bovine serum and added to the apical chamber of transwells. Media from the basal compartment was sampled after 60 min and the number of beads present in the sample was analyzed by flow cytometry using an LSR II flow cytometer (BD).

## TriCEPS screen

For initial conjugation of TriCEPS to protein, EsxA or transferrin (300 µg) dissolved in 150 µL 25 mM HEPES pH 8.2 buffer was added to 1.5 µL of the TriCEPS reagent (Dualsystems Biotech) and incubated at 20°C for 90 min with gentle agitation. During this time, $6 \times 10^8$ Caco-2 cells were detached from tissue culture plates using 10 mM EDTA in PBS. Cells were split into three aliquots, cooled to 4°C, and pelleted. Each pellet was resuspended in PBS pH 6.5 and sodium metaperiodate was added to a final concentration of 1.5 mM in order to gently oxidize the cell surface. Cells were then incubated with sodium metaperiodate in the dark for 15 min at 4°C. Cells were washed twice with PBS pH 6.5 and split into two new aliquots. TriCEPS coupled EsxA was added to one aliquot and TriCEPS coupled transferrin was added to the other aliquot and incubated for 90 min at 4°C with gentle agitation. Samples were then washed, lysed via sonication, and digested with trypsin. The TriCEPS reagent:ligand:receptor complex was then affinity purified and samples were analyzed using a Thermo LTQ Orbitrap XL spectrometer fitted with an electrospray ion source. Samples were measured in data-dependent acquisition mode in a 90 min gradient using a 10 cm C18 packed column. Samples were analyzed with a statistical ANOVA model with p-values adjusted to control the experiment-wide false discovery rate (FDR). The adjusted p-value obtained for each protein was plotted against the fold enrichment between the two experimental conditions. The area in the volcano plot with an enrichment factor of 4 fold or greater and an FDR-adjusted p-value less than or equal to 0.01 was defined as the receptor candidate space.

## Immunoprecipitation

EsxA was expressed and purified as described above. EsxA or PBS alone was then biotinylated by the Sulfo-SBED reagent (Thermo 33033) according to the manufacturer instructions and excess reagent was removed using PD-10 columns (GE Healthcare 17-0851-01). $1 \times 10^7$ HBE cells were detached from tissue culture plates using 10 mM EDTA in PBS. Cells were washed, resuspended in HBSS, and incubated with 1.5 µM EsxA or with PBS alone for 2 hr at 4°C with gentle agitation. Cells were then washed and exposed to UV light for 30 min at room temperature to allow for covalent cross-linking. Cells were lysed with RIPA buffer and lysate was incubated with streptavidin-conjugated magnetic beads (Thermo 88816). Proteins were eluted by boiling and analyzed by SDS-PAGE followed by western blotting with rabbit anti-SR-B1 antibody (Abcam 52629; RRID:AB_882458).

## Mouse NALT/human adenoid immunofluorescence

Mouse NALT sections were obtained as previously described (*Nair et al., 2016*). Briefly, mouse NALT (after decalcification) and human adenoid specimens were embedded in paraffin, sectioned (5 µm), and mounted on glass slides. Slides were deparaffinized using xylene and ethanol washes followed by heat mediated antigen-retrieval in 10 mM sodium citrate (pH 6.0). Endogenous peroxidase activity was quenched and slides were blocked in 10% donkey serum in PBS for 3 hr at room temperature. Slides were washed with PBS and incubated with a 1:100 dilution of mouse NKM 16-2-4 and rabbit anti-SR-B1 in 2% donkey serum in PBS overnight at 4°C. Slides were then washed with PBS and incubated with a 1:500 dilution of AlexaFluor 568 conjugated goat-anti-mouse secondary antibody (Thermo A-11004; RRID:AB_2534072) or with HRP-conjugated donkey-anti-rabbit secondary

antibody (Thermo A16023; RRID:AB_2534697) in 2% donkey serum in PBS for 1 hr at room temperature. Slides were then washed with PBS and incubated with Cy5 tyramide (Perkin Elmer SAT705A001EA) for 8 min. Slides were then washed with PBS, incubated with DAPI, washed with PBS, mounted in Prolong Gold antifade reagent, and imaged using an AxioImager MN microscope (Zeiss). At least three fields of view were imaged per independent experiment.

### Mouse intranasal infection

Mtb Erdman and all mutants were grown in 7H9 and 0.05% Tween-80 until mid-log phase. Cultures were washed three times with PBS, centrifuged to remove clumps, and sonicated to yield a single-cell suspension. Bacteria were resuspended to yield a final concentration of $1 \times 10^8$ bacteria in 10 µL PBS. BALB/c mice obtained from The Jackson Laboratory (RRID:IMSR_JAX:000651) were infected with 10 µL of the bacterial suspension intranasally. NALT from three to five mice were collected, homogenized, and plated on 7H11 (Difco 283810) plates supplemented with 10% OADC to enumerate the number of bacteria deposited on Day 0. Mice were sacrificed on Day 7 post-infection and cervical lymph nodes were collected, homogenized, and plated on 7H11 plates. Plates were incubated in a 37°C incubator for 3 weeks to allow for colony formation.

### Adenoid culture and infection

Adenoid samples were obtained from children undergoing elective adenoidectomy for obstructive sleep apnea. Excised adenoids were immediately placed in DMEM, subsequently dissected into 3–4 pieces depending on the size of the adenoid, weighed, and mounted in a 2% agar pad such that only the mucosal surface was exposed. The adenoid pieces were then incubated overnight at 37°C in DMEM supplemented with 20% fetal bovine serum, 2 mM L-glutamine, 1% sodium pyruvate, 1% non-essential amino acids, 1 mM HEPES, 50 ug/mL kanamycin, and 50 ug/mL ampicillin to kill commensal bacteria. The following morning, liquid cultures of GFP Mtb (Kanamycin-resistant) grown to mid-log phase were washed three times with PBS and centrifuged and sonicated to remove clumps. Bacteria were then diluted to $1 \times 10^7$ bacteria/mL and 1 mL of inoculum was added to the adenoid and incubated at 37°C for 1 hr. Adenoids were then washed, minced into small pieces, and pushed through a 100 µm nylon cell strainer (Corning 431752). Cells were centrifuged, washed in ACK (Ammonium-Chloride-Potassium) lysis buffer (Gibco A10492-01), and then resuspended in FACS buffer (PBS + 2% FBS). Cells were stained with a 1:100 dilution of mouse anti-EpCAM Brilliant Violet 421 (Biolegend 324219; RRID:AB_11124342), mouse PE-NKM-16-2-4 (Miltenyi 130-102-150; RRID:AB_2660295), or rabbit anti-SR-B1 in FACS buffer, washed, and then incubated with a 1:500 dilution of AlexaFluor 488 conjugated donkey-anti-rabbit secondary antibody (Thermo R37118; RRID:AB_2556546). Cells were washed and fixed in 4% paraformaldehyde for 1 hr followed by counting on an LSRII flow cytometer and analyzed using FlowJo software.

### Statistical analysis

Statistical analysis was performed using GraphPad Prism (RRID:SCR_002798). For in vitro transwell infections to determine bacterial binding or translocation, one-way ANOVA with corrections for multiple comparisons was performed. For in vitro determination of antibody staining, one-way ANOVA with corrections for multiple comparisons was performed. For in vivo adenoid infections or receptor expression, the paired non-parametric Wilcoxon matched pairs signed rank test was performed. For in vivo mouse infections and determination of CFU, the non-parametric Mann-Whitney U test was performed.

## Acknowledgements

We thank Beth Levine and members of the Shiloh Lab for constructive feedback on the manuscript. This work is supported by the Burroughs Wellcome Fund 1017894 (MUS), Welch Foundation I-1964–20180324 (MUS), NIH U01 AI125939-04 (MUS), NIH U19 AI142784-01 (MUS), NIH 5T32AI005284-40 (HSK), and NIH R01 HL131597-03 (PWS).

## Additional information

### Funding

| Funder | Grant reference number | Author |
| --- | --- | --- |
| National Institute of Allergy and Infectious Diseases | AI125939 | Michael U Shiloh |
| National Institute of Allergy and Infectious Diseases | AI142784 | Michael U Shiloh |
| National Institute of Allergy and Infectious Diseases | 5T32AI005284 | Haaris S Khan |
| National Heart, Lung, and Blood Institute | HK131597 | Philip W Shaul |
| Burroughs Wellcome Fund | 1017894 | Michael U Shiloh |
| Welch Foundation | I-1964-20180324 | Michael U Shiloh |

The funders had no role in study design, data collection and interpretation, or the decision to submit the work for publication.

### Author contributions

Haaris S Khan, Conceptualization, Formal analysis, Investigation, Visualization, Writing - original draft, Project administration, Writing - review and editing; Vidhya R Nair, Conceptualization, Formal analysis, Investigation, Writing - review and editing; Cody R Ruhl, Samuel Alvarez-Arguedas, Jorge L Galvan Rendiz, Luis H Franco, Investigation, Writing - review and editing; Linzhang Huang, Ron B Mitchell, Resources, Writing - review and editing; Philip W Shaul, Resources, Funding acquisition, Writing - review and editing; Jiwoong Kim, Data curation, Software, Formal analysis, Writing - review and editing; Yang Xie, Resources, Software, Supervision, Writing - review and editing; Michael U Shiloh, Conceptualization, Formal analysis, Supervision, Funding acquisition, Visualization, Writing - original draft, Project administration, Writing - review and editing

### Author ORCIDs

Michael U Shiloh (iD) https://orcid.org/0000-0003-4329-2253

### Ethics

Human subjects: Human adenoids were obtained from children undergoing elective adenoidectomy for sleep apnea after informed consent was obtained from parents or guardians. This study was reviewed by the University of Texas Southwestern Institutional Review Board (protocol STU 062016-087).

Animal experimentation: Animal experiments were reviewed and approved by the Institutional Animal Care and Use Committee at the University of Texas Southwestern (protocol 2017-101836) and followed the eighth edition of the Guide for the Care and Use of Laboratory Animals. The University of Texas Southwestern is accredited by the American Association for Accreditation of Laboratory Animal Care (AAALAC).

### Decision letter and Author response

Decision letter https://doi.org/10.7554/eLife.52551.sa1
Author response https://doi.org/10.7554/eLife.52551.sa2

## Additional files

### Supplementary files

• Supplementary file 1. Variant table for WT Mtb, Mtb *eccD1*::Tn and MtbΔ*esxA*. Genomic DNA from the indicated strains was sequenced and variants identified by comparison to the Mtb Erdman reference genome.

- Supplementary file 2. Mass spectrometry peptide analysis of transferrin or EsxA cell surface binding. TriCEPS-labeled transferrin or EsxA were affinity purified after binding to the surface of Caco-2 cells, and bound proteins analyzed by mass spectrometry.
- Supplementary file 3. Key resources table.
- Transparent reporting form

## Data availability

All data generated or analyzed during this study are included in the manuscript and supporting files. Whole genome sequencing data have been deposited at NCBI Sequence Read Archive, Accession PRJNA605439. All materials are available upon request.

The following dataset was generated:

| Author(s) | Year | Dataset title | Dataset URL | Database and Identifier |
|---|---|---|---|---|
| Shiloh MU | 2020 | Identification of scavenger receptor B1 as the airway microfold cell receptor for Mycobacterium tuberculosis | https://www.ncbi.nlm.nih.gov/bioproject/PRJNA605439/ | NCBI BioProject, PRJNA605439 |

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
