## [Decision Letter]

**Acceptance summary:**

Thank you for your detailed response to the reviewers' comments and for submitting this interesting manuscript to *eLife*! We are excited about the findings you present showing that Mtb EsxA is involved in Mtb invasion of M cells through its engagement with the SR-B1 airway M cell receptor. These findings are an important advance in our understanding of *Mycobacterium tuberculosis* invasion at different host sites, where the role and mechanisms of M cell invasion is understudied and poorly understood.

**Decision letter after peer review:**

Thank you for submitting your article "Identification of scavenger receptor B1 as the airway microfold cell receptor for *Mycobacterium tuberculosis*" for consideration by *eLife*. Your article has been reviewed by three peer reviewers, one of whom is a member of our Board of Reviewing Editors, and the evaluation has been overseen by Wendy Garrett as the Senior Editor. The following individuals involved in review of your submission have agreed to reveal their identity: Patricia Champion (Reviewer #2).

The reviewers have discussed the reviews with one another and the Reviewing Editor. Everyone is very positive about the manuscript and have drafted this decision to help you prepare a revised submission.

Summary:

In this well-presented and interesting manuscript, the authors demonstrate a role for Mtb EsxA in the invasion of M cells through it's engagement with the SR-B1 airway M cell receptor for Mtb. The study is a nice extension of their previous work and an important advance in our understanding of *Mycobacterium tuberculosis* infection. The reviewers request the following revisions to support the data interpretations made by the authors.

Essential revisions:

1) Please add the following controls to the data shown:

A) Genetic complementation of any of the Mtb strains lacking the T7SS genes.

B) Add EsxB as a control protein for Figure 2 D-H.

C) Add the uninfected control transwell data to Figure 1G.

D) Add a loading control and the EsxA protein pulled down for the western blot in Figure 3B. In addition, are the molecular weight markers correctly indicated for this figure and are there more marker sizes? Unglycosylated SR-BI is indeed ~55-57 kDa, however, this does not seem to be the case in this blot based on the markers. Furthermore, a fully glycosylated SR-BI should be no more than 82 kDa. As such, glycosylation should be verified by EndoH treatment. It is also possible that the band at 130 kDa may represent an SR-BI oligomer, but this should be verified as the estimated molecular weight does not match that of an SR-BI dimer. Have the authors checked to see if that 130 kDa band is actually SR-BI potentially bound to another protein?

2) Please add the following missing information/data:

A) Coomassie gels of the purification of EsxA and EsxB and a discussion of any quality control performed on the proteins, for example MS analysis, to confirm identity/ purity.

B) Methods regarding the LC-MS analysis.

C) For all microscopic images, please indicate how many overall images the presented ones in the manuscript represent. (i.e. How many fields of view were imaged per independent experiment?).

D) Quantification of colocalization of EsxA and SRB-1 in Figure 4.

E) Figure 1B shows approximately 10 bacteria in the field, but the graph in C shows 100-150 area of bacteria per field. Is bacteria area different from the number of bacteria? How were the numbers that are graphed acquired?

F) Figure 2E and 3D state a similar number of nuclei per field was observed, but Figure 2D shows at least two fold higher nuclei than 3C. How are these numbers determined/calculated?

G) For Figure 1B and others, the authors should present the brightfield image to determine where the cell boundaries are to try and distinguish bacteria adherent or inside host cells.

H) In Figure 3, the enrichment of APOE and SRB1 peptides by EsxA IP are significantly less than the enrichment of the transferrin receptor and transferrin. How much or each "receptor" did you pull down relative to the amount of EsxA and transferrin? Is the volcano plot the result of one, two or three independent replicates from the IP?

I) Is it possible to include a schematic of the transwell assay in Figure 1?

J) For Figure 4: The molecular weight of the SR-BI band should be indicated. Is a lower exposure of this immunoblot available?

3) Data files and statistics revisions:

A) The complete proteomics data file supporting the volcano plot in Figure 3A should be submitted.

B) The statistical methods that led to the p-values used to generate the volcano plot in Figure 3A should be added.

C) It is unclear that the Student's t-test is appropriate throughout. For example, I think the t-test is appropriate for comparing treated vs untreated (as RajiB vs Control for a single strain), but inappropriate to go between two strains, or proteins (like 1C-E and 2A, C).

D) Please either designate whether it should be assumed that conditions not called out in the figures are not significantly different from each other, or add a supplemental table of the outcome of all statistical comparisons. For example, would be interesting to know if the deccD +/- RajiB is significantly different from each other in 1C and 1E.

4) Please add the following reference related revisions:

A) The correct reference for SR-BI glycosylation (subsection “Scavenger receptor class B type 1 binds EsxA and is expressed on M cells in vitro”) should be used (older Krieger papers).

B) "Two of the most abundant T7SS secreted proteins are EsxA and EsxB" - Please add reference.

C) Discussion paragraph four – please indicate that Shen et al. is a review. Otherwise, provide the original reference for the identification of SR-BI as an HDL receptor.

Some other points to consider that would strengthen the conclusions made:

1) Something to consider - is performing a direct IP between EsxA and soluble domains of SRB-1 possible? This would sure up the conclusion that SRB1 is an EsxA "receptor" and that the interaction between the two is direct. Alternatively, if the crosslinking assay you are using demonstrates a direct interaction between EsxA and SRB1, despite the fact that the pull down is from lysates, this needs to be clearly stated and discussed.

2) The authors' findings raise the question of whether chemical inhibition of SR-B1 could inhibit binding of EsxA and translocation of Mtb. Using a chemical inhibitor of SR-B1 would alleviate any concerns that genetic knockdown of SR-B1 affects M cell biology more generally. A chemical inhibitor could also be used in vivo.

3) Is it possible to isolate primary M cells from SR-BI^-/-^ mice to definitively identify SR-BI as an EsxA receptor? This would increase the significance of these studies.

4) There are several pieces of data in the manuscript that indicate that EsxA may not be the only factor, or that there may be additional receptors for EsxA on M cells. For example, Figure 4: There are still EsxA+ cells in the absence of SRB1. Also, is translocation into M cells higher for beads with EsxA than Mtb? Does this imply that Mtb surface might include additional factors that modulate this process? Do M. tb without esxA still translocate at a low% , again indicating additional factors? This possibility should be clearly addressed in the Discussion.

---

## [Author Response]

Essential revisions:1) Please add the following controls to the data shown:A) Genetic complementation of any of the Mtb strains lacking the T7SS genes.

We thank the reviewer for highlighting the need for complementation which has historically been used to control for polar effects or off-target mutations. In our manuscript we chose to use two independent mutants that each have a non-functional T7SS, as it is highly unlikely that both strains would share similar off-target effects of the introduced genetic modifications beyond the known broad impact on the T7SS encoded in the RD1 locus, or that both would have spontaneous mutations in other genomic regions. In addition, both of the strains we used in this study, the MtbΔ*esxA* (Stanley et al., 2007) and Mtb *eccD1::Tn5370* (Stanley et al., 2007; Stanley et al., 2003) have been studied previously, including with complementation for T7SS activity (i.e. EsxA secretion), thus excluding polar effects and major genomic changes in the original mutants. To confirm that there were no additional mutations that might have developed during laboratory passage, we sequenced the genomes of all three strains, Mtb Erdman, Mtb ErdmanΔ*esxA* and Mtb Erdman *eccD1::Tn5370* and compared them to the published Mtb Erdman reference genome (GenBank: AP012340). The whole genome sequencing identified the expected deletion in *esxA* along with the presence of the residual transposon sequences and hygromycin cassette used in the generation of the mutant(Mtb *esxA* in the reference genome is ERDMAN_RS20430 and the deletion is located at NC_020559.1:4333535-4333786) and the transposon insertion in *eccD1* (Mtb *eccD1* in the reference genome is ERDMAN_RS20440 and the insertion is at NC_020559.1:4337340). As expected, all three strains shared minor differences with the reference genome, and importantly, the Mtb Erdman WT and Mtb *eccD1::Tn5370* were identical to each other. When we compared the Mtb Erdman WT to the Mtb ErdmanΔ*esxA,* there was only a single coding variant encoding a missense mutation in Mtb ErdmanΔ*esxA* compared to the Mtb Erdman WT in a putative oxidoreductase. We have uploaded the sequencing data to the Sequence Read Archive (SRA submission # PRJNA605439), and include a variant table in the supplemental material (Supplementary File 1).

B) Add EsxB as a control protein for Figure 2 D-H.

This experiment is now reflected in revised Figure 2D-H.

C) Add the uninfected control transwell data to Figure 1G.

We did not have uninfected data to add, but we did include the translocation data for the control transwells not co-cultured with Raji-B cells in new Figure 1H and Figure 1I.

D) Add a loading control and the EsxA protein pulled down for the western blot in Figure 3B.

We added a loading control as suggested. For the EsxA pulldown, we attempted to detect EsxA after the biotin switch assay but were unable to detect the protein with either the monoclonal EsxA antibody or the anti-His antibody. It is possible that the reaction impacted the epitopes detected by the antibodies.

In addition, are the molecular weight markers correctly indicated for this figure and are there more marker sizes? Unglycosylated SR-BI is indeed ~55-57 kDa, however, this does not seem to be the case in this blot based on the markers. Furthermore, a fully glycosylated SR-BI should be no more than 82 kDa. As such, glycosylation should be verified by EndoH treatment. It is also possible that the band at 130 kDa may represent an SR-BI oligomer, but this should be verified as the estimated molecular weight does not match that of an SR-BI dimer. Have the authors checked to see if that 130 kDa band is actually SR-BI potentially bound to another protein?

We thank the reviewer for identifying the issue regarding the molecular weight of SR-B1. Regarding the molecular weight discrepancy of SR-B1 (130 kDa vs 82 kDa), we returned to our Western blot/IP data and discovered that the reason for the discrepancy was the primary antibody we used to detect SR-B1. In the experiment performed, we used a polyclonal anti-rabbit antibody against SR-B1 (Novus Biologicals NB400-104), which for unclear reasons demonstrated SR-B1 at ~130 kDa both in our hands and on the product website. When we used the mouse monoclonal antibody against SR-B1 that is more commonly used in the field (abcam; ab52629), we demonstrated SR-B1 at the expected molecular weight of 82 kDa (see Author response image 1). We have now redone the biotin switch assay using the ab52629 monoclonal antibody, obtained the appropriate size for SR-B1 and have revised the figure accordingly (New Figure 3B).

2) Please add the following missing information/data:A) Coomassie gels of the purification of EsxA and EsxB and a discussion of any quality control performed on the proteins, for example MS analysis, to confirm identity/ purity.

A Coomassie gel is now provided in the supplemental materials (New Figure 2—figure supplement 1).

B) Methods regarding the LC-MS analysis.

Detailed methods for LC-MS are now included in the Materials and methods section.

C) For all microscopic images, please indicate how many overall images the presented ones in the manuscript represent. (i.e. How many fields of view were imaged per independent experiment?).

Depending on the experiment, between 3 to 5 fields per condition were imaged and quantified. This information has been added to the Materials and methods section.

D) Quantification of colocalization of EsxA and SRB-1 in Figure 4.

Quantification is presented in Figure 4F.

E) Figure 1B shows approximately 10 bacteria in the field, but the graph in C shows 100-150 area of bacteria per field. Is bacteria area different from the number of bacteria? How were the numbers that are graphed acquired?

We thank the reviewer for highlighting this confusing point. The image in Figure 1B is a high-power representative image, but not of an entire field. We initially chose to quantify bacteria by using ImageJ to determine the pixel intensity per field as an unbiased surrogate for manually counting bacteria. For clarity, we have changed the analysis so that number of bacteria per 100 cells is quantified in the revised manuscript.

F) Figure 2E and 3D state a similar number of nuclei per field was observed, but Figure 2D shows at least two fold higher nuclei than 3C. How are these numbers determined/calculated?

As above, the images presented are representative images, not of the field in its entirety. Nuclei per field were counted using ImageJ.

G) For Figure 1B and others, the authors should present the brightfield image to determine where the cell boundaries are to try and distinguish bacteria adherent or inside host cells.

While we appreciate the reviewer’s concerns, brightfield images cannot distinguish between adherent and internalized bacteria. To accomplish this would require confocal microscopy with Z-stacks, and such experiments are quite challenging for transwells. As we were interested in binding to the cells, which could include surface binding and subsequently internalized bacteria, we did not pursue additional experimentation.

H) In Figure 3, the enrichment of APOE and SRB1 peptides by EsxA IP are significantly less than the enrichment of the transferrin receptor and transferrin. How much or each "receptor" did you pull down relative to the amount of EsxA and transferrin? Is the volcano plot the result of one, two or three independent replicates from the IP?

The transferrin receptor is a ubiquitously expressed and abundant cell surface protein, which is why the company we contracted, Dualsystems Biotech, provides transferrin as the positive (experimental) control. We only present the transferrin/transferrin receptor data to demonstrate the success of the TriCEPS assay, not for comparative purposes. In addition, it is not possible to quantify the amount of each receptor pulled down using this method. The volcano plot is the result of one experiment with three independent biologic replicates. The latter point is now clearly explained in the figure legend to Figure 3A.

I) Is it possible to include a schematic of the transwell assay in Figure 1?

Added to Figure 1 as new Figure 1B.

J) For Figure 4: The molecular weight of the SR-BI band should be indicated. Is a lower exposure of this immunoblot available?

The molecular weight has now been added and a lower exposure image now replaces Figure 3B.

3) Data files and statistics revisions:A) The complete proteomics data file supporting the volcano plot in Figure 3A should be submitted.

The complete proteomics data file has been added as supplementary data (new Supplementary File 2).

B) The statistical methods that led to the p-values used to generate the volcano plot in Figure 3A should be added.

Added to the Materials and methods section as suggested.

C) It is unclear that the Student's t-test is appropriate throughout. For example, I think the t-test is appropriate for comparing treated vs untreated (as RajiB vs Control for a single strain), but inappropriate to go between two strains, or proteins (like 1C-E and 2A, C).

For the entire manuscript, we replaced Student’s t-test by one-way ANOVA with corrections for multiple comparisons as it is a more rigorous statistical analysis tool. The Materials and methods and figure legends have been updated with this change.

D) Please either designate whether it should be assumed that conditions not called out in the figures are not significantly different from each other, or add a supplemental table of the outcome of all statistical comparisons. For example, would be interesting to know if the deccD +/- RajiB is significantly different from each other in 1C and 1E.

We have added a comment to the legend for each figure where appropriate indicating that if comparisons are not directly highlighted, that they are not statistically significant.

4) Please add the following reference related revisions:A) The correct reference for SR-BI glycosylation (subsection “Scavenger receptor class B type 1 binds EsxA and is expressed on M cells in vitro”) should be used (older Krieger papers).

Thank you for the suggestion. We revised this section to reflect the new data (see comment 1D above) and include the original paper by Acton et al. to indicate the correct MW.

B) "Two of the most abundant T7SS secreted proteins are EsxA and EsxB" - Please add reference.

Thank you for the suggestion. We have included references detailing the abundance of these two proteins.

C) Discussion paragraph four - please indicate that Shen et al. is a review. Otherwise, provide the original reference for the identification of SR-BI as an HDL receptor.

We changed the reference to indicate the original identification of SR-B1 as an HDL receptor.

Some other points to consider that would strengthen the conclusions made:1) Something to consider - is performing a direct IP between EsxA and soluble domains of SRB-1 possible? This would sure up the conclusion that SRB1 is an EsxA "receptor" and that the interaction between the two is direct. Alternatively, if the crosslinking assay you are using demonstrates a direct interaction between EsxA and SRB1, despite the fact that the pull down is from lysates, this needs to be clearly stated and discussed.

Thank you for the suggestion. Performing an IP of EsxA with a soluble SR-B1 domain would be a challenge as it is not known if the three-dimensional structures of soluble (i.e. recombinant) and transmembrane SR-B1 are the same, and thus soluble SR-B1 may not bind EsxA for technical rather than biologic reasons. We performed two independent assays using EsxA to probe for cell surface binding. While both included a cross-linking step (TriCEPs and biotin switch), they were also both performed on whole cells, not lysates, and both demonstrated an interaction between EsxA and SR-B1.

2) The authors' findings raise the question of whether chemical inhibition of SR-B1 could inhibit binding of EsxA and translocation of Mtb. Using a chemical inhibitor of SR-B1 would alleviate any concerns that genetic knockdown of SR-B1 affects M cell biology more generally. A chemical inhibitor could also be used in vivo.

Thank you for the suggestion. While we recognize this is an important experiment, we are concerned that an inhibitor of SR-B1 activity that blocks HDL internalization (such as the established HDL internalization inhibitor BLT-1) might not share the same inhibitory activity for Mtb binding/internalization. As such, only a positive result (i.e. an inhibitor successfully prevents Mtb internalization) would be interpretable. In addition, administration of such an inhibitor in vivo could yield confounding results because it could not only impact M cell internalization but also macrophage SR-B1 mediated internalization.

3) Is it possible to isolate primary M cells from SR-BI^-/-^ mice to definitively identify SR-BI as an EsxA receptor? This would increase the significance of these studies.

SR-B1^-/-^ mice are nearly impossible to breed and have multiple developmental and physiologic abnormalities. Ongoing work in the lab is focused on developing a M cell specific Cre mouse that will allow for cell type specific conditional deletion of SR-B1 from M cells by crossing to an SR-B1^fl/fl^ mouse. However, validating and testing Cre expression, crossing these mice and then testing them for M cell activity is beyond the scope of this manuscript.

4) There are several pieces of data in the manuscript that indicate that EsxA may not be the only factor, or that there may be additional receptors for EsxA on M cells. For example, Figure 4: There are still EsxA+ cells in the absence of SRB1. Also, is translocation into M cells higher for beads with EsxA than Mtb? Does this imply that Mtb surface might include additional factors that modulate this process? Do M. tb without esxA still translocate at a low% , again indicating additional factors? This possibility should be clearly addressed in the Discussion.

Thank you for raising these important points. We have expanded the Discussion to incorporate the idea that there may be additional EsxA receptors on M cells, and that other Mtb factors may also mediate M cell entry.

References:

Stanley, SA, Johndrow, JE, Manzanillo, P, Cox, JS.2007. The Type I IFN response to infection with *Mycobacterium tuberculosis* requires ESX-1-mediated secretion and contributes to pathogenesis. J Immunol *178*:3143–3152.